# VISUALLY-AUGMENTED LANGUAGE MODELING

**Weizhi Wang**[†], **Li Dong**[‡], **Hao Cheng**[‡], **Haoyu Song**[‡],
**Xiaodong Liu**[‡], **Xifeng Yan**[†], **Jianfeng Gao**[‡], **Furu Wei**[‡]
[†]University of California, Santa Barbara   [‡]Microsoft Research
weizhiwang@ucsb.edu, {lidong1, haocheng}@microsoft.com

## ABSTRACT

Human language is grounded on multimodal knowledge including visual knowledge like colors, sizes, and shapes. However, current large-scale pre-trained language models rely on text-only self-supervised training with massive text data, which precludes them from utilizing relevant visual information when necessary. To address this, we propose a novel pre-training framework, named VALM, to **V**isually-**a**ugment text tokens with retrieved relevant images for **L**anguage **M**odeling. Specifically, VALM builds on a novel latent text-image alignment method via an image retrieval module to fetch corresponding images given a textual context. With the visually-augmented context, VALM uses a visual knowledge fusion layer to enable multimodal grounded language modeling by attending to both text context and visual knowledge in images. We evaluate VALM on various visual knowledge-intensive commonsense reasoning tasks, which require visual information to excel. The experimental results illustrate that VALM outperforms all strong language-only and vision-language baselines with substantial gains in reasoning object commonsense including color, size, and shape. Our code is available at https://github.com/Victorwz/VaLM.

## 1 INTRODUCTION

Large-scale pre-trained language models (PLMs) have achieved great success in promoting state of the art on various natural language understanding and generation tasks (Devlin et al., 2019; Radford et al., 2019; Liu et al., 2019; Yang et al., 2019; Brown et al., 2020; Wang et al., 2022). PLM self-supervision training largely benefits from harvesting local context information in the pre-training corpus. To further strengthen such contextual self-supervision, recent seminal works, e.g. GPT-3 (Brown et al., 2020) and Megatron-LM (Narayanan et al., 2021), focus on increasing the model size and the scale of pre-training corpus. With billions of parameters, these tremendous PLMs exhibit incredible ability as zero-shot or few-shot learners. More remarkably, PLMs can achieve human-parity performance on various downstream tasks, even without any task-specific supervision. Another major research line of PLMs is to enhance the language model with auxiliary knowledge (Wei et al., 2021), including entity knowledge (Yu et al., 2020), relational knowledge (Zhang et al., 2019; Qin et al., 2021), text chunk (Lewis et al., 2020; Wu et al., 2022; Borgeaud et al., 2021), etc. The incorporation of various knowledge resources to PLMs mitigates the drawbacks of *local* contextual attention, bringing additional relevant *global* context that benefits both language understanding and generation tasks.

Since current unimodal PLMs lack visual knowledge grounding, they inevitably suffer from the *hallucination* problem, which refers to the inconsistent or false statements generated by PLMs with respect to the world knowledge (Logan et al., 2019). For instance, the PLMs may predict the color of the sky as red only due to the statistical contextual correlations between the token "color" and "red" in the pre-training corpus, neglecting the commonsense facts.

In this paper, we propose a novel framework to enable language model pre-training to take full advantage of both local text context and corresponding visual knowledge. Recent work on joint vision-language model (VLM) pre-training (Su et al., 2020; Tan & Bansal, 2020) relies on *explicit* alignments between text and image, e.g. supervised image captioning data, which limits the cross-modality fusion during fine-tuning/inference over text without accompanying images. As a consequence, later in our experiments (section 3), those prominent VLMs are found to achieve unsatisfactory performance on visual knowledge-intensive commonsense reasoning tasks. Instead, we design a flexible text-image

alignment mechanism via an image retrieval module that gathers related images for each token as visual augmentation. To achieve better language-vision grounding, we propose a visual knowledge fusion layer to enable joint attention across visually-augmented context including both textual tokens and retrieved images. Based on this, we build up a **V**isually-**a**ugmented **L**anguage **M**odel, VALM, with flexible on-the-fly visual knowledge enhancement.

We evaluate the effectiveness of the proposed VALM on various commonsense reasoning and language-only benchmarks. Experimental results demonstrate that our model consistently outperforms the unimodal and multimodal baselines in terms of object commonsense reasoning. Remarkably, our method substantially improves +14.50%, +17.80%, and +11.68% accuracy on MEMORYCOLOR, RELATIVESIZE and OBJECTSHAPE datasets, respectively. Additional experiments on natural language understanding tasks also validate that the proposed visually-augmented language modeling framework could be helpful to improve the fundamental natural language understanding capability of PLMs.

Our contributions are summarized as follows:

- We propose a novel visually-augmented casual language model, VALM, to enable the language model to utilize visual knowledge flexibly and effectively. Through the proposed visual knowledge fused language modeling, VALM is capable of accomplishing tasks with the high demand of cross-modality knowledge, such as visual commonsense reasoning.

- We design a framework to construct flexible on-the-fly text-image alignments and fuse augmented images into the context of language modeling. We implement an image retrieval module to query token-level representation in a large-scale cached image database and retrieve its nearest neighbors as the augmentation. With the proposed visual knowledge fusion layer, VALM can effectively take full advantage of both language information from local text context and visual information from retrieved images.

- Experimental results demonstrate that VALM effectively alleviates the hallucination problem of PLMs via introducing visual knowledge in language model pre-training. VALM achieves significant performance improvements in inferring the commonsense object properties.

## 2 METHODS

We propose a novel multi-modal pre-trained language model, which is augmented with retrieved images, named VALM. The architecture of VALM is presented in Figure 1. VALM augments each token in pre-training text corpus with $k$ retrieved related images. VALM uses an image retrieval module to retrieve corresponding images for each token. The image retrieval module deploys a pre-trained CLIP model, which is capable of unifying the textual query and image candidates into a joint embedding space. VALM constructs a cached large-scale image knowledge base using image encoder of CLIP, and uses the contextual representation of each token as textual query to search its nearest neighbors in image knowledge base. With the help of the unified text and image embedding space provided by CLIP, the image nearest neighbors are taken as augmented images of each token to construct text and image alignments. We then propose a visual-knowledge fusion layer to enable learned hidden state to attend to both texts and augmented images.

### 2.1 VALM: VISUALLY-AUGMENTED LANGUAGE MODELING

Given an input text sequence $\{\mathbf{x}_i\}_{i=1}^N$, the embedding layer first encodes input vector $\{\mathbf{x}_i\}_{i=1}^N$ into embedding space and outputs the initial hidden state $\mathbf{H}^0$ to the successive Transformer decoder layers. Then the proposed VALM model encodes $\mathbf{H}^0$ into visual knowledge fused contextual representations at difference levels $\mathbf{H} = \{\mathbf{H}^l\}_{l=1}^L$ via $L-1$ Transformer decoder layers and one special visual knowledge fusion layer. Each Transformer decoder layer is identical to Vaswani et al. (2017), which outputs the contextual representations at different semantic levels given the representation from the previous layer $\mathbf{H}^l = \text{Layer}_l(\mathbf{H}^{l-1}), l \in [1, L]$.

The visual knowledge fusion layer is proposed as a variant of the Transformer decoder layer to incorporate visual knowledge in contextual learning via joint attention on both text contexts and augmented images. The visual knowledge fusion layer is injected in the second-to-last layer of VALM. The visual knowledge is stored in corresponding augmented image representations, obtained from

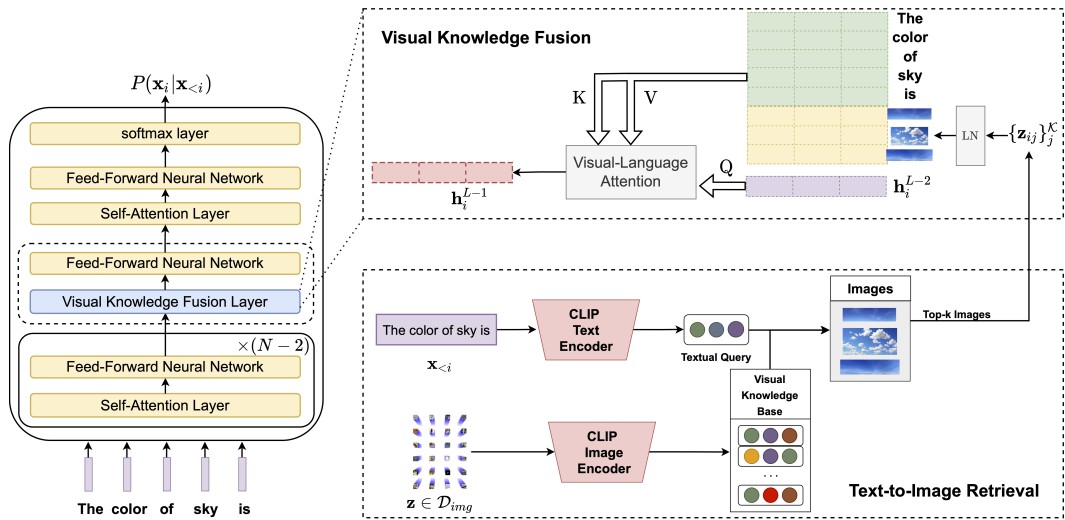

Figure 1: Overview of visually-augmented language modeling (VALM). We conduct dense retrieval to get top-$k$ images for the input context at each time step. Then the visual knowledge fusion layer attends to both text tokens and retrieved images. The vision-language fused representation is fed back to Transformer for language modeling.

image retrieval module $\{\{\mathbf{z}_{ij}\}_{j=1}^{\mathcal{K}}\} = f_{rt}(\mathbf{x}_i)$. Then the visual knowledge fusion layer takes the input including both contextual representation of the previous layer and augmented image sets and outputs a visual-knowledge fused contextual representation $\mathbf{H}^{L-1} = \text{VisualLayer}(\{\mathbf{H}_i^{L-2}, \{\mathbf{z}_{ij}\}_{j=1}^{\mathcal{K}}\}_{i=1}^{N})$. Finally, the output contextual representations are passed into the output projection layer and a *softmax* function is used to compute the token probability $P(\mathbf{x}_i|\mathbf{x}_1, \cdots, \mathbf{x}_{i-1}) = \text{softmax}(W\mathbf{H}^L + b)$.

We conduct *generative unsupervised pre-training* (Radford et al., 2019) for VALM on a large-scale text corpus. The training objective of VALM is the standard left-to-right language modeling objective, which maximizes the likelihood of the next word token based on the left context:

$$\max \sum_{x \in \mathcal{D}} \sum_{i=1}^{|\mathbf{x}|} \log P(\mathbf{x}_i|\mathbf{x}_1, \cdots, \mathbf{x}_{i-1}), \tag{1}$$

where $\mathbf{x}$ represents a sentence randomly sampled from the large-scale pre-training text corpus $\mathcal{D}$.

## 2.2 IMAGE RETRIEVAL

The visual knowledge corresponding to a specific token is stored in its correlated images. Therefore, to prepare the fused visual knowledge, VALM deploys an image retrieval module to retrieve augmented images, denoted as $f_{rt}(\cdot)$. In order to achieve multi-modality text-image retrieval, it is of great importance to building up a discriminator to assess the correlation of every image in the extremely large-scale open image knowledge bases to the specific text representation. CLIP (Radford et al., 2021) proposed a simple-yet-effective method to connect images and texts into a unified multi-modal embedding space. We directly deploy the pre-trained CLIP model to encode the images and texts to enable a nearest neighbor text-image retrieval. Specifically, the pre-trained CLIP model we use in constructing the image retrieval module includes a ResNet-50x16 (He et al., 2016) model as an image encoder and a Transformer (Vaswani et al., 2017) model as a text encoder. Here, we only use the CLIP model as the backbone of our image retrieval module, and the CLIP parameters are not updated during the pre-training process of VALM.

**Image Knowledge Base Creation.** The image knowledge base of the retrieval module is the cache of a set of image keys, which are the high-level visual representations of images. Given an image $\mathbf{z} \in \mathcal{D}_{img}$, such visual representation can be obtained via forwarding image $\mathbf{z}$ to the pre-trained CLIP image encoder. Then the whole image knowledge base ($\mathcal{Z}$) is constructed by taking the output hidden state $f_{\theta_I}(\mathbf{x})$ as image keys: $\mathcal{Z} = \bigcup_{\mathbf{z} \in \mathcal{D}_{img}} \{f_{\theta_I}(\mathbf{z})\}$, where $\theta_I$ represents the image encoder parameters.

**Textual Query.**   We take the contextual representation of each token as the query in the nearest neighbor search. For each sentence $\mathbf{x} \in \mathcal{D}$, the contextual representation of $i$-th token is computed via $f_{\theta_T}(\mathbf{x}_{<i})$, where $\theta_T$ represents the text encoder parameters. As the input sequence length of VALM generally exceeds the input length limitation of 75 tokens of CLIP text encoder, the long context $\mathbf{x}_{<i}$ is cut off into a context-chunk $\mathbf{y}_i$ for fitting in CLIP text encoder: $\mathbf{y}_i = \begin{cases} \mathbf{x}_{[t,i-1]}, & i - t < 75, \\ \mathbf{x}_{[i-75,i-1]}, & i - t \geq 75, \end{cases}$ where $t$ is the index of the closest stop character before $i$-th token. Then the textual query for $i$-th token is computed as its context-chunk representation as $f_{\theta_T}(\mathbf{y}_i)$.

**$k$NN Text-Image Retrieval.**   The retrieval module uses the contextual representation to search the cached image knowledge base ($\mathcal{Z}$) and retrieves $k$ nearest neighbor image keys w.r.t. dot product distance. As the pre-trained CLIP model has learned a joint embedding space for text and image domain, the retrieved images $\{\mathbf{z}_{ij}\}_{j=1}^{\mathcal{K}}$ are thus regarded as the top-$k$ relevant images to the query.

## 2.3   VISUAL KNOWLEDGE FUSION

With the help of the image retrieval module, each token in the pre-training corpus is augmented with $k$ corresponding images, and these augmented images are represented in the joint embedding space with texts. Then the augmented image representations are directly treated as auxiliary "context" in the learning process.

As the conventional Transformer decoder layer uses the multi-head self-attention (Vaswani et al., 2017) to learn the contextual representation, we extend it to a joint-attention mechanism and propose a novel visual knowledge fusion layer to enable each token to attend to both contexts and retrieval images jointly. In addition, due to the inconsistency in magnitude and distribution between contextual hidden states and retrieved image representations, we apply Layer Normalization (Ba et al., 2016) on retrieved $\mathcal{K}$ image representations to alleviate such inconsistency, denoted as $\mathrm{LN}_{img}$. Assume that the hidden state output for $i$-th token is $\mathbf{h}_i$ and the corresponding retrieved images are $\{\mathbf{z}_{ij}\}_{j=1}^{\mathcal{K}}$, the hidden state $\mathbf{H}_i^{L-1}$ is computed as:

$$\mathbf{Q} = \mathbf{H}^{L-2}W^Q + b^Q, \mathbf{K} = \mathbf{H}^{L-2}W^K + b^K, \mathbf{V} = \mathbf{H}^{L-2}W^V + b^V, \tag{2}$$

$$\dot{\mathbf{k}}_{ik} = \mathrm{LN}_{img}(\mathbf{z}_{ik})W^K + b^K_{img}, \dot{\mathbf{v}}_{ik} = \mathrm{LN}_{img}(\mathbf{z}_{ik})W^V + b^V_{img}, \tag{3}$$

$$e_i = \frac{\mathbf{Q}_i \mathbf{K}^T}{\sqrt{d}}, a_i = \frac{\exp(e_i)}{\sum_{j=1}^{\mathcal{L}} \exp(e_{ij}) + \sum_{k=1}^{\mathcal{K}} \exp(e_{ik})}, \tag{4}$$

$$e_{ik} = \frac{\mathbf{Q}_i \dot{\mathbf{k}}_{ik}^T}{\sqrt{d}}, a_{ik} = \frac{\exp(e_{ik})}{\sum_{j=1}^{\mathcal{L}} \exp(e_{ij}) + \sum_{k=1}^{\mathcal{K}} \exp(e_{ik})}, \tag{5}$$

$$\mathbf{H}_i^{L-1} = a_i\mathbf{V} + \sum_k a_{ik}\dot{\mathbf{v}}_{ik}, \tag{6}$$

where $\mathbf{Q}_i, \dot{\mathbf{k}}_{ik}, \dot{\mathbf{v}}_{ik} \in \mathcal{R}^{\mathrm{E}}$, $\mathbf{K}, \mathbf{V} \in \mathcal{R}^{|\mathbf{x}| \times \mathrm{E}}$, $e_i, a_i \in \mathcal{R}^{|\mathbf{x}|}$. The hidden state output from the previous layer $\mathbf{H_i^{L-1}}$ is linearly projected into contextual queries, keys, and values $\mathbf{Q}, \mathbf{K}, \mathbf{V}$ separately. $\mathcal{K}$ is the number of retrieved images for each token, and E is the embedding dimension for both context and image representations. In order to generate image-specific attention keys and values, we adopt image-specific bias $b^K_{img}, b^V_{img}$ in linear projections and reuse the contextual projection weights $W^K, W^V$ to generate image-specific attention keys and values. Moreover, it is vital to mention that the image-specific attention keys and values are distinct for each query token, which is highly different from self-attention where the contextual keys and values are kept the same for each token. A secondary subscript $k$ is used to denote different image representations for the $i$-th token.

## 3   EXPERIMENTS

## 3.1   PRETRAINING SETUP

**Text Corpus.**   We use the English corpus of CC-100 (Conneau et al., 2020) as the pre-training text corpus for both VALM and baseline GPT-2[*]. CC-100 corpus is one of the largest high-quality

| Task | Example Prompt | Object / Pair | Answer |
|---|---|---|---|
| Object Color Reasoning | *The color of* `[object]` *is* `[answer]` | *the sky* | *blue* |
| Object Shape Reasoning | *The shape of* `[object]` *is* `[answer]` | *apple* | *round* |
| Object Size Reasoning | *Is* `[Item1]` *larger than* `[Item2]? ` `[answer]` | *(sofa, cat)* | *Yes* |

Table 1: Evaluation examples of object color, shape, and size reasoning.

web-crawl text data. The English monolingual dataset of CC-100 contains about 55 billion tokens, stored in 301 GiBs disk storage. Due to the limitation of computing resources, we only consume 15% of CC-100 English monolingual corpus for pre-training VALM and baseline GPT-2*.

**Image Data.** We use the LAION Open Image Dataset (Schuhmann et al., 2021) as the image knowledge base for dense retrieval. To the best of our knowledge, the LAION Open Dataset is one of the world's largest openly available image-text-pair dataset with 400 million samples. Due to the disk space limitation, we randomly select half of LAION images for the dense text-image retrieval, which is 200M images in total.

**Pre-training Hyperparameters.** The proposed model deploys transformer decoder architecture with 124M trainable parameters. Hyperparameter setting and training details are presented in Appendix B.1.

**Retrieval Module.** For the implementation of the dense text-image retrieval module, we use the `faiss` (Johnson et al., 2021) toolkit to construct the efficient index. The `faiss` index contains the whole 200M image keys and provides the efficient nearest neighbor search. For efficiency purposes, we quantize all image keys to 32 bytes. `faiss` index stores image keys in clusters to speed up the search, which requires the additional training process to learn the cluster centroids. We use 10M keys for learning 131k cluster centroids and search 32 clusters to find the nearest neighbors during inference. We load the `faiss` index to GPU to achieve efficient dense text-image retrieval.

## 3.2 VISUAL KNOWLEDGE INTENSIVE TASKS

The visual information stored in retrieved images can play a useful role in providing relevant visual knowledge to help language models perform better grounded commonsense reasoning. Such helpful visual information can be colors, positions, sizes, spatial relations, etc. The task of object commonsense reasoning requires language models to predict the correct visual property for a given object. To excel these tasks typically require models to capture and utilize intensive visual knowledge without any explicit text demonstrations or external knowledge bases. Due to reporting biases, such descriptive text of object properties rarely appears in text corpora, likely making this type of knowledge absent from language models. Thus, those visual knowledge-intensive tasks are likely challenging for both language models and vision-language models.

We first compared VALM and recent baselines on four object commonsense reasoning datasets, MEMORYCOLOR (Norlund et al., 2021), COLORTERMS (Bruni et al., 2012), OBJECTSHAPE (Zhang et al., 2022a) reasoning, and RELATIVESIZE (Bagherinezhad et al., 2016). In addition, we use another physical interaction question answering dataset (PIQA) (Bisk et al., 2020), to evaluate whether such visual commonsense knowledge could be implicitly encoded and utilized in the question answering process. In Table 1, we provide examples for different visual commonsense reasoning tasks.

**MEMORYCOLOR and COLORTERMS Dataset.** The *memory color* of a concrete object is the typical color an object appears in, e.g. the color of banana is mostly memorized as yellow. Norlund et al. (2021) proposed this dataset for evaluating visual knowledge transfer in multi-modal language models. The dataset contains 109 objects paired with their memory color labels, an illustrating picture, and a descriptor. The COLORTERMS dataset also contains a list of common items manually labeled with their commonsense color. Both datasets hold a set of 11 color labels.

**OBJECTSHAPE Dataset.** Zhang et al. (2022a) proposed a visual commonsense dataset with a set of object attributes like shape. The dataset of object shapes contains 140 objects with their shape label. The OBJECTSHAPE dataset consists of 12 shape categories.

**RELATIVESIZE Dataset.** Bagherinezhad et al. (2016) proposed the RELATIVESIZE dataset, which includes a total of 486 object pairs between 41 physical objects. The task of object size reasoning requires the model to predict the size relations between two given objects, e.g., an ant is smaller than an elephant. The size information is again rarely included and described in text, while it is much easier to capture from the images. We convert the size relation reasoning task into a binary question-answering form with "Yes"/"No" answers.

**PHYSICAL INTERACTION QUESTION ANSWERING.** Physical Interaction Question Answering (PIQA) is proposed and designed to investigate the physical commonsense knowledge of existing language models (Bisk et al., 2020). Completing such question answering tasks requires the language model to effectively utilize physical commonsense knowledge, i.e. knowledge of basic properties of the objects (flexibility, curvature, and being porous). Language models are supposed to first achieve the perception of objects and later encode such physical knowledge into the language modeling process. Each data sample in PIQA contains one goal and two solutions. The model is supposed to figure out and predict the more reasonable and appropriate solution between two candidates.

**Evaluation Setting.** We evaluate VALM and all baseline methods in a zero-shot manner without any task-specific tuning. Specifically, VALM takes the input consisting of textual prompts and objects during inference and predicts the property label as the last token. The prompts used in evaluating object color, shape, and size reasoning performance are listed in Appendix Table 11. We use the top-1 accuracy as the evaluation metric and compute the average accuracy of all listed prompts to increase evaluation robustness. For PIQA, we follow Shwartz et al. (2020) to use the cross-entropy loss as the scorer for each potential solution $score(s_{ij}) = CE([g_i, s_{ij}]), j \in [0, 1]$. Then the solution with lower scores is selected as the prediction. The classification accuracy is used as the evaluation metric.

**Baselines.** We consider both pretrained language-only and vision-language models as baselines. In particular, three strong language models are considered for comparison with VALM, including 1) GPT-2* (Radford et al., 2019); 2) BERT Devlin et al. (2019); and 3) CaptionBERT (Zhang et al., 2022a), a pre-trained auto-encoding language model on Oscar's (Li et al., 2020) caption-based text data. Here, GPT-2* is re-implemented and trained from scratch using the identical training data, hyper-parameter settings, and model size as the proposed VALM. Additionally, we also compare VALM with prominent vision-language models, including 1) OSCAR (Li et al., 2020), a pre-trained vision-language model with learned representations that capture channel-invariant factors (i.e. object tags) at the semantic level; 2) VisualBERT (Li et al., 2019), a vision-language model with learned joint contextualized representations across vision and language; 3) CLIP (Radford et al., 2021), a vision-language system with one image encoder and one text encoder which are mapped into a same cross-modal embedding space. We directly use OSCAR and VisualBERT as auto-encoding language models for zero-shot evaluations. For CLIP, we first retrieve the corresponding image using the concatenated query prompt and the given object. Then, the dot-product similarity of the retrieved image vector and the candidate-aware text vector (including the query prompt, the given object, and one candidate label) is used to rank. Finally, the top-ranked candidate label is regarded as the prediction for evaluation.

**Results.** The main results on four object commonsense reasoning datasets are summarized in Table 2. The two variants of VALM ($\mathcal{K} = 4, 8$) significantly outperform all considered language models and vision-language models on object color and shape reasoning datasets, with an improvement of +14.50%, +13.56%, and +11.68% on MEMORYCOLOR, COLORTERMS, and OBJECTSHAPE respectively. Moreover, the proposed VALM with $\mathcal{K} = 4$ achieves an encouraging result with +17.80% accuracy gain over the best baseline, VisualBERT on RELATIVESIZE. The substantial improvements on these datasets demonstrate that VALM takes full advantage of visual knowledge (object visual property) to complete the corresponding visual commonsense reasoning. Surprisingly, the zero-shot evaluation results of all auto-encoding language models and vision-language models are below 40% accuracy on object color and shape reasoning datasets. Although pretrained with aligned text-image pairs, those vision-language models cannot effectively utilize relevant visual knowledge from their jointly contextualized vision-language representations. Among language models, the auto-regressive PLMs significantly outperform auto-encoding PLMs, suggesting that auto-regressive PLMs are likely better at zero-shot reasoners. We also observe that retrieving more images for each token results in a performance drop for object size and shape reasoning. We attribute the degradation

| Model | $\mathcal{K}$ | Color (ACC↑) | | Shape (ACC↑) | Size (ACC↑) |
|-------|-----|------------|------------|-------------|-------------|
| | | MEMORYCOLOR | COLORTERMS | OBJECTSHAPE | RELATIVESIZE |
| GPT-2* | N/A | 44.14% | 39.10% | 51.09% | 47.22% |
| BERT | N/A | 24.34% | 26.33% | 31.86% | 34.78% |
| CaptionBERT | N/A | 24.84% | 28.40% | 38.14% | 66.05% |
| CLIP | N/A | 26.25% | 23.08% | 13.66% | 47.99% |
| OSCAR | N/A | 20.32% | 16.86% | 33.14% | 50.14% |
| VisualBERT | N/A | 26.68% | 38.02% | 11.14% | 67.23% |
| VALM | 4 | 53.99% | **52.66%** | **62.77%** | **85.03%** |
| VALM | 8 | **58.64%** | 50.19% | 59.41% | 62.35% |

Table 2: Accuracy on object commonsense reasoning datasets. GPT-2* is the re-implemented model with identical pre-training data and hyper-parameter settings to VALM. $\mathcal{K}$ represents the number of images augmented to each token. Best performance is marked with bold.

| Model | GPT-2* | BERT | CaptionBERT | VALM ($\mathcal{K}$=4) | VALM ($\mathcal{K}$=8) |
|-------|--------|------|-------------|----------------|----------------|
| PIQA (ACC↑) | 62.53% | 54.73% | 53.97% | 64.31% | **64.64%** |

Table 3: Accuracy on Physical-Interaction Question-Answering benchmark.

to the increased noise brought by augmenting with more images which causes model confusion when differentiating relevant visual information from irrelevant one.

PIQA is a more challenging task that requires the model to reason useful implicit object properties and utilize these commonsense in the question answering process. The results on PIQA are presented in Table 3. As is shown, VALM outperforms all baseline language models with +2.11% accuracy improvement. The two variants of VALM achieve almost identical performance because the selection for the correct solution is based on the language modeling perplexity, indicating that the two variants demonstrate similar language modeling capability.

## 3.3 NATURAL LANGUAGE UNDERSTANDING AND LANGUAGE MODELING TASKS

The casual language modeling pre-training task enables PLMs to naturally perform natural language understanding (NLU) and long-text modeling. Therefore, the zero-shot natural language understanding and language modeling performance are widely adopted to evaluate the capability of PLMs (Radford et al., 2019). Here, we evaluate VALM and the most relevant language model baseline GPT-2* on four NLU datasets, SST-2 (Socher et al., 2013), MPQA (Wiebe et al., 2005), DBPeida (Auer et al., 2007), and AGNews (Zhang et al., 2015). The prediction accuracy is used as the evaluation metric. In addition, following Radford et al. (2019), Wikitext-103 (Merity et al., 2017) and Lambda corpus (Paperno et al., 2016) are considered to study the language modeling performance in a zero-shot manner. We report perplexity for two corpora and also report last-word prediction accuracy for Lambada corpus.

The results on natural language understanding are summarized in Table 4. It is easy to see that VALM achieves decent improvements on all four NLU tasks, indicating that the cross-modality knowledge learned in our model is likely helpful for typical natural language understanding. Thus, our visually-augmented language modeling framework can be further explored to enhance the natural language understanding ability of PLMs. Table 5 illustrates the results of language modeling tasks. Again, VALM slightly improves the perplexity on both datasets, +0.68 on Wikitext-103 and +0.08 on Lambda. A similar trend is observed for the final word prediction accuracy on Lambada. Different from previous visual knowledge intensive commonsense reasoning tasks (subsection 3.2), we find that VALM models with different numbers of retrieved images ($\mathcal{K} = 8$ vs $\mathcal{K} = 4$) perform similarly on the intrinsic language modeling task, suggesting that VALM can effectively ignore irrelevant visual information when the task is unlikely to benefit from visual knowledge. In other words, visual commonsense reasoning tasks require more fine-grained fusions of text and image, i.e. locating the text object in the image set, extracting relevant vision information, and verbalizing reasoning output. In contrast, a certain portion of text from general language modeling corpora s is probably not visually related. Thus, only a coarse-grained fusion is sufficient here (e.g. deciding if the image set is

| Model | $\mathcal{K}$ | SST-2 ACC↑ | MPQA ACC↑ | DBPedia ACC↑ | AGNews ACC↑ |
|---|---|---|---|---|---|
| Majority | N/A | 50.90% | 50.00% | 9.4% | 25.0% |
| GPT-2* | N/A | 68.04% | 71.25% | 67.20% | 53.51% |
| VALM | 4 | 70.12% | 78.70% | 72.27% | 53.81% |
| VALM | 8 | 67.33% | 77.35% | 68.48% | 59.77% |

Table 4: Zero-shot evaluation results on natural language understanding tasks (SST-2, MPQA, DBPedia, AGNews). Majority: majority class.

| Model | $\mathcal{K}$ | Wikitext-103 PPL↓ | Lambda PPL↓ | Lambda ACC↑ |
|---|---|---|---|---|
| GPT-2* | N/A | 36.44 | 42.46 | 42.17% |
| VALM | 4 | 35.78 | 42.51 | 42.65% |
| VALM | 8 | 35.76 | 42.38 | 42.87% |

Table 5: Zero-shot evaluation results on language modeling tasks. We report perplexity (PPL) on Wikitext-103 and Lambada and final word prediction accuracy (ACC) on Lambada.

useful), making the language modeling evaluation less affected by the retrieval noise from augmented images.

## 3.4 ABLATION STUDIES

So far, we empirically verify the effectiveness and superiority of VALM in utilizing visual knowledge for both visual knowledge-intensive tasks and traditional language understanding and modeling. To figure out how the visual information takes effect in our model, we focus on two questions here: 1) Is the model capable of using the retrieved image representations as "auxiliary" contexts? What is the effect of disabling such retrieved image representations during inference? To evaluate this, we design an ablation model which set $\mathcal{K} = 0$ and disables image retrieval and fusion during inference. 2) Does the model learn to leverage visual knowledge in the retrieved images? What is the effect of directly augmenting randomly-retrieved image representations during inference? Thus, an ablation model which retrieves random images as augmentations during inference is used for probing. The results of the two ablation models, Randomly-Retrieval and Disable-Retrieval during the inference stage, are listed in the first two rows of Table 6. As we can see, both changes to the image retrieval result in noticeable performance degradation on all evaluation tasks. In particular, we find that disabling the image retrieval and augmenting no image during inference also makes a huge difference to the language modeling perplexity on two corpora, which is more related to pure text corpus rather than augmented images. Therefore, it suggests that VALM is able to effectively capture rich semantics from both pretraining sources, i.e. text corpus as well as augmented images. In other words, the improved zero-shot task transferability of VALM relies on visual information from augmented images, which complements the linguistic knowledge learned via text-based self-supervised learning. The results of the Randomly-Retrieval ablation model further illustrate that achieving the capability of integrating visual knowledge cannot be realized by only augmenting unrelated images to language models, while only context-relevant images can make a true difference.

VALM proposes a novel visual knowledge fusion layer with a joint context-image attention mechanism as a key component to fuse visual knowledge into the language model. The separate linear projection layers are regarded as important components to map contexts into different embedding spaces for attention keys and values. Therefore, the proposed joint self-attention mechanism naturally holds three variations to generate image keys and values: establish image-specific linear projections, reuse contextual linear projections, and only use specific linear bias for augmented images. We conduct the ablation study to evaluate the effect of these three alternatives on image linear projections. The results in Table 6 demonstrate that adopting image-specific projection bias outperforms directly sharing the contextual projection bias. Introducing additional image-specific linear projection weights does not lead to further performance improvement. Thus, we take the strategy of only adding additional linear bias for augmented images and reuse contextual linear weights in generating visual attention keys and values for implementation convenience and parameter efficiency.

| Ablation Model | $\mathcal{K}$ | MemoryColor ACC↑ | ColorTerms ACC↑ | RelativeSize ACC↑ | Wikitext-103 PPL↓ | Lambada PPL↓ | Lambada ACC↑ |
|---|---|---|---|---|---|---|---|
| Randomly-Retrieval | 4 | 41.48% | 38.46% | 49.63% | 37.89 | 43.56 | 41.35% |
| Disable-Retrieval | 0 | 43.12% | 40.53% | 59.36% | 39.22 | 44.59 | 41.20% |
| $W_{img}^{K,V}, b_{img}^{K,V}$ | 4 | 48.62% | 41.86% | 83.54% | 35.95 | 42.28 | 42.67% |
| $W^{K,V}, b^{K,V}$ | 4 | 47.85% | 46.30% | 62.04% | 35.95 | 42.33 | 41.74% |
| $W^{K,V}, b_{img}^{K,V}$ (VALM) | 4 | 53.99% | 52.66% | 85.03% | 35.78 | 42.51 | 42.65% |

Table 6: Ablation studies on the effects of Randomly-Retrieval and Disabling-Retrieval during inference stage (Upper part). Second ablation study on the effects of introducing extra image-specific attention key and value projection weights $W_{img}^{K,V}$ or bias $b_{img}^{K,V}$ in Equation 3 for augmented images. The proposed model VALM is shown in the last row which introduces only image-specific bias and reuses contextual weight in attention key and value projection layers.

## 4 RELATED WORK

**Pre-trained Language Models.** Pre-trained language models (PLMs) revolutionized NLP research. Enabled by attention mechanism (Bahdanau et al., 2015) and Transformer architecture (Vaswani et al., 2017), the state-of-the-art PLMs, including BERT (Liu et al., 2019), GPT (Radford et al., 2018; 2019), RoBERTa (Liu et al., 2019), ELECTRA (Clark et al., 2020), T5 (Raffel et al., 2020), and OPT (Zhang et al., 2022b), have become the dominant approach in NLP tasks via the paradigm of pre-training on large-scale text corpora and fine-tuning on downstream tasks. With the exponential scaling up of model size, a surprising fact emerged that the PLMs like GPT-3 (Brown et al., 2020) can work as few-shot or zero-shot learners.

**Vision-Language Models.** Vision-language tasks are at the intersection area of both modalities of vision and language, like visual-question answering (Agrawal et al., 2015), and image captioning (Chen et al., 2015). ViL-BERT (Lu et al., 2019) firstly proposed to generate image region features via object detection and then learn joint multi-modal representations via an interacted two-stream model. OSCAR (Li et al., 2020) proposed to introduce object tags detected in images as anchor points to solve the issue of high demand for image-text alignments. Another significant pathway for VLMs is to construct a unified embedding space for texts and images and use textual prompts to extract task-specific labels during inference, of which the representative models are CLIP (Radford et al., 2021) and ALIGN (Jia et al., 2021).

**Visually-Grounded Language Learning.** Visually-grounded language learning is an emerging research topic in vision-language learning, in which the proposed VALM can be categorized in this area with other prior works like Vokenization (Tan & Bansal, 2020), VidLanKD (Tang et al., 2021), and iACE (Lu et al., 2022). Visual information and knowledge can be memorized by the PLMs via fusion layer or concatenated inputs. However, extracting and utilizing the visual information efficiently and effectively is still difficult for uni-modal language models. Vokenization concatenated tokens and token-related images as "vokens", transferring sentence-level caption text to token-level "voken" with a Vokenizer model.

## 5 CONCLUSION

In this paper, we propose a multi-modal framework VALM to enable auto-regressive language modeling to effectively utilize visual knowledge. Specifically, an effective text-to-image retrieval module is designed to construct latent text-image alignments for visually-grounded language modeling. Empowered by pre-training, VALM achieves improved zero-shot task transfer on downstream tasks. Experiments on various visual knowledge-intensive tasks demonstrate the effectiveness of our model over recent vision-language models. VALM also achieves decent improvements over language models on multiple representative natural language understanding tasks. For future work, we plan to adapt the model architecture to encoder-only and encoder-decoder Transformer backbones. We are also interested in more input modalities for VALM.

## ACKNOWLEDGEMENTS

We would like to thank the anonymous reviewers for the helpful comments. We appreciate Zewen Chi and Hangbo Bao for the fruitful discussions, and Yaru Hao for helpful suggestions on evaluation benchmarks.

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

# A  ADDITIONAL RESULTS

## A.1  TIME-COST EFFECTS OF RETRIEVAL AND IMAGESET SIZE

Introducing efficient image retrieval on GPU brings a linear increase in inference time cost (about 2.1 times of text-only GPT-2* baseline), shown in Table 7. This cost is negligible with larger-size language models because the model forward cost will increase many times while the retrieval cost will not change with the model size. The retrieval cost can be further improved by searching fewer clusters or decreasing the number of encoding bytes for approximate image keys, with a minor trade-off on the performance. Moreover, efficient nearest neighbor search is an active research area (Guo et al., 2020) and we could try other efficient search tools to accelerate the retrieval.

As the introduced retrieval time cost is proportional to the size of imageset for dense retrieval, we provide more details on the relationship between retrieval time cost and imageset size, presented in Table 7. Concluded from Table 7, there is no significant performance decrease with the smaller imageset size from the original 200M down to 10M. As the 10M set is still large and sufficient for providing enough visual knowledge, we will consider deploying a 10M size imageset to train VALM for potential real-time industry applications.

| Image Size | Color (ACC↑) | | Shape (ACC↑) | Size (ACC↑) | Timecost |
| | MEMORYCOLOR | COLORTERMS | OBJECTSHAPE | RELATIVESIZE | (GPT2* as 1x) |
| --- | --- | --- | --- | --- | --- |
| 200M | 53.99% | 52.66% | 62.77% | 85.03% | 2.06x |
| 100M | 53.50% | 49.71% | 61.39% | 81.84% | 1.91x |
| 10M | 51.79% | 47.49% | 62.18% | 82.15% | 1.79x |
| 1M | 51.87% | 46.31% | 48.51% | 82.35% | 1.74x |

Table 7: Accuracy on object commonsense reasoning datasets of VALM ($\mathcal{K} = 4$) with variants of retrieval imageset size. $\mathcal{K}$ represents the number of images augmented to each token.

## A.2  COMPARISONS WITH ADDITIONAL STRONG BASELINES

We compare VALM with Vokenization (Tan & Bansal, 2020) on four visual-knowledge-intensive tasks, and the results are shown in Table 8. In addition, we evaluate the performance of large language models on the visual–knowledge-intensive tasks for stronger and more fair comparisons. We evaluate the OPT (1.3B parameters) (Zhang et al., 2022b) model on these visual–knowledge-intensive tasks and the results are presented in Table 8. VALM(124M parameters) significantly outperforms the OPT-1.3B on four datasets, which further demonstrates the challenge of solving those visual-knowledge-intensive tasks and the effectiveness of our method.

| Model | $\mathcal{K}$ | Color (ACC↑) | | Shape (ACC↑) | Size (ACC↑) |
| | | MEMORYCOLOR | COLORTERMS | OBJECTSHAPE | RELATIVESIZE |
| --- | --- | --- | --- | --- | --- |
| OPT-1.3B | N/A | 39.25% | 41.03% | 19.21% | 50.78% |
| Vokenization | N/A | 14.18% | 19.97% | 48.35% | 43.28% |
| VALM | 4 | 53.99% | **52.66%** | **62.77%** | **85.03%** |
| VALM | 8 | **58.64%** | 50.19% | 59.41% | 62.35% |

Table 8: Accuracy on object commonsense reasoning datasets. $\mathcal{K}$ represents the number of images augmented to each token. Best performance is marked with bold.

## A.3  SCALING EFFECT OF VALM

We train the 355M model (GPT-2 Medium Size) of VALM (k=8) to evaluate the effects of scaling up model parameters. The results are presented in Table 9 and the model performance is significantly improved on four visual knowledge-intensive datasets. We will seek more computation resources to train large size VALM models.

| Model | $\mathcal{K}$ | Color (ACC↑) | | Shape (ACC↑) | Size (ACC↑) |
|---|---|---|---|---|---|
| | | MEMORYCOLOR | COLORTERMS | OBJECTSHAPE | RELATIVESIZE |
| VALM-124M | 8 | 58.64% | 50.19% | 59.41% | 62.35% |
| VALM-355M | 8 | 65.82% | 55.36% | 70.0% | 72.79% |

Table 9: Accuracy on object commonsense reasoning datasets. $\mathcal{K}$ represents the number of images augmented to each token. Best performance is marked with bold.

## A.4 ABLATION STUDY OF $\mathcal{K}$

We further conduct another ablation study by setting the number of augmented images $\mathcal{K} = 1$ for VALM, which is very similar to the CLIP (Radford et al., 2021) inference. The results are presented in Table 10. VALM (k=1) significantly outperforms CLIP in all visual-knowledge-intensive tasks, validating the effectiveness of our method.

| Model | $\mathcal{K}$ | Color (ACC↑) | | Shape (ACC↑) | Size (ACC↑) |
|---|---|---|---|---|---|
| | | MEMORYCOLOR | COLORTERMS | OBJECTSHAPE | RELATIVESIZE |
| CLIP | N/A | 26.25% | 23.08% | 13.66% | 47.99% |
| VALM | 1 | 52.43% | 46.01% | 64.55% | 86.73% |

Table 10: Accuracy on object commonsense reasoning datasets. $\mathcal{K}$ represents the number of images augmented to each token.

## A.5 CASE STUDIES

We provide a case study in the object color reasoning task for VALM. In order to reason the correct commonsense color of objects sky and parsley, VALM takes the input combination of the prompt and the object as "the color of [object] is". Then we present the retrieval results of top-4 corresponding images to the textual query in Figure 2.

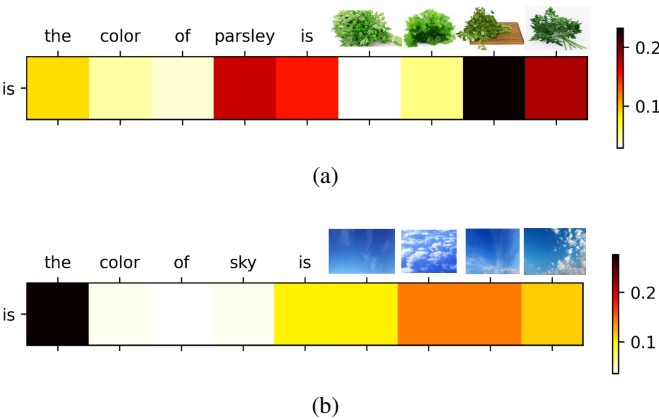

(a)

(b)

Figure 2: The attention matrix visualization given the query prompt "the color of [object] is" for VALM. VALM achieves accurate image retrieval of top-4 images corresponding to the objects of sky and parsley as augmented images, shown in the horizontal index of each subfigure.

## A.6 COLORIZATION EFFECT

We conduct another interesting ablation case study to evaluate the effect of image color changes in the object color reasoning task. Specifically, VALM predicts the color label of an apple as red based on the commonsense in both contexts and retrieved images. The original prediction probability distribution is presented in *Blue Bars* of Figure 3(b). Then we replace the retrieved images with $\mathcal{K}$ unusual images of green apples in OBJECTCOLORIZATION dataset (Anwar et al., 2020), shown in

Figure 3(a). The predicted probability distribution for 11 color types given replaced colorization objects is presented in *Orange Bars* of Figure 3(b). We could observe a clear probability increase in the color type of green and a decrease in that of red, which is confronted with the colorization process. This ablation study demonstrates VALM is capable of extracting useful visual knowledge from retrieved object images and inferring correct semantics based on that. Given retrieved object images in different colors, VALM could extract the correct color knowledge and adopt it in its semantic inference stage.

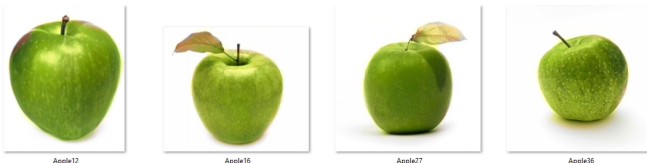

(a) Images for green apples in OBJECTCOLORIZATION dataset as replacements.

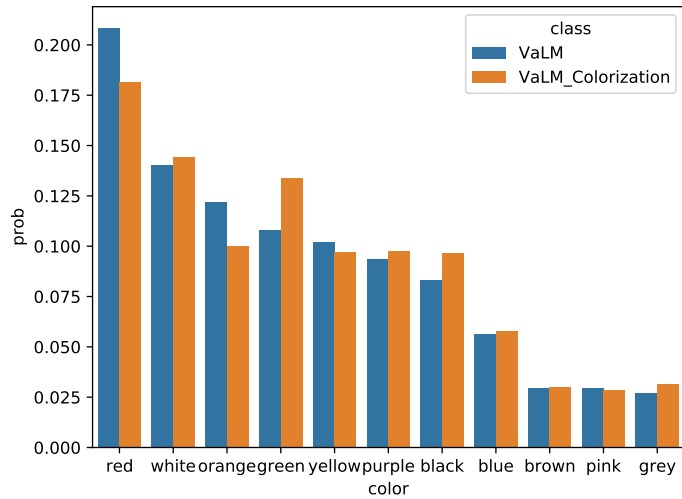

(b) Probability Distribution Visualization for retrieved images and colorization images.

Figure 3: The visualization of the predicted probability distribution on 11 object color types with retrieved images and colorization images, respectively. The adopted prompt for reasoning the object color of an apple is "the color of [object] is".

## B    EXPERIMENTAL DETAILS

### B.1    PRE-TRAINING HYPERPARAMETERS AND TRAINING DETAILS

The implementation of models and all experiments are based on the `fairseq` (Ott et al., 2019) toolkit. The proposed model deploys transformer decoder architecture with 124M trainable parameters, in which $n_{\text{layer}} = 12, n_{\text{head}} = 12, d_{\text{embed}} = 768$. We deploy Adam (Kingma & Ba, 2015) ($\beta_1 = 0.9, \beta_2 = 0.98$) optimizer and train all models with $lr = 0.0005, t_{\text{warmup}} = 4000, \text{dropout} = 0.1, bsz = 128, len = 512$. The layer normalization over the retrieved image keys is initialized with $\epsilon$ of 0.00001. VALM reuses the identical lower-cased byte pair encoding (BPE) (Sennrich et al., 2016) representation with a 49152 vocab size of CLIP text encoder. The proposed VALM and re-implemented GPT-2[*] are trained for 500k steps using 16 Nvidia Tesla V100-SXM2-32GB GPUs. The encoded 200M image knowledge base takes up 274GiBs disk storage and the trained *faiss* approximate retrieval index takes another 14GiBs storage.

### B.2    PROBE TEMPLATES

We present all zero-shot query prompts and labels for 4 object commonsense reasoning datasets and 4 natural language understanding benchmarks in Tabele 11.

| Task | Prompt | Labels |
|---|---|---|
| **Object Color Reasoning** | Q: What is the color of [DESCRIPTOR] [ITEM]? A: It is [Label]
Q: What is the colour of [DESCRIPTOR] [ITEM] ? A: It is [Label]
What is the color of [DESCRIPTOR] [ITEM]? It is [Label]
What is the colour of [DESCRIPTOR] [ITEM]? [Label]
The color of [DESCRIPTOR] [ITEM] is [Label]
The usual color of [DESCRIPTOR] [ITEM] is [Label]
[DESCRIPTOR] [ITEM] usually has the color of [Label]
What is the usual color of [DESCRIPTOR] [ITEM]? [Label]
What is the typical color of [DESCRIPTOR] [ITEM]? [Label] | {red, white, orange, green, blue, yellow, purple, black, pink, grey, brown} |
| **Object Shape Reasoning** | Is [ITEMA] larger than [ITEMB]? [Label]
[ITEM] can be shape [Label]
[ITEM] has shape [Label]
[ITEM] is of shape [Label]
The shape of [ITEM] can be [Label]
The shape of the [ITEM] is [Label] | {cross, heart, octagon, oval, polygon, rectangle, rhombus, round, semicircle, square, star, triangle} |
| **Object Size Reasoning** | Is [ITEMA] larger than [ITEMB]? [Label]
Is [ITEMA] taller than [ITEMB]? [Label]
Is [ITEMA] higher than [ITEMB]? [Label]
[ITEMA] is larger than [ITEMB], is it true? [Label]
[ITEMA] is taller than [ITEMB], is it true? [Label]
[ITEMA] is larger than [ITEMB], is it true? [Label] | {Yes, No} |
| **SST-2**
**MPQA**
**DBPedia**
**AGNews** | Review: [Sentence] Sentiment: [Label]
Review: [Sentence] Sentiment: [Label]
input: [Sentence] type: [Label]
input: [Sentence] type: [Label] | {Positive, Negative}
{Positive, Negative}
{company, school, ..., book}
{world, ..., technology} |

Table 11: The prompts and prediction labels used to query the model predictions on the zero-shot evaluation of 4 object commonsense reasoning and 4 natural language understanding benchmarks. The labels for DBPedia are {company, school, artist, athlete, politics, transportation, building, nature, village, animal, plant, album, film, book} and the labels for AGNews are {world, sports, business, technology}.

