# OpenReview forum: "Visually-Augmented Language Modeling"
_ICLR.cc/2023/Conference — ICLR 2023 poster_

### Official Review · Reviewer_j6aD · 2022-10-25

**Confidence:** 4
**Correctness:** 3
**Technical Novelty And Significance:** 3
**Empirical Novelty And Significance:** 3
**Recommendation:** 6

**Clarity, Quality, Novelty And Reproducibility:**

The writing is clear. There is some novelty, but more explanations are needed to separate it from other publications. The paper has enough details to reproduce the results, though I think the reproduction would require many computation resources and time. So it would be great to have the pretrained models also released by the authors.

**Strength And Weaknesses:**

Strength:

This work proposes a new architecture for augmenting language modeling with auxiliary visual cues. The authors tested multiple evaluation datasets for their proposed and other candidate models. The improvement in these datasets compared to the other models tested in this work is significant. The authors also show some ablation studies that the proposed image retrieval module helps the performance.

Weakness:

The biggest worry I have is about the evaluation in this work, which is critical to be resolved before I can fully back the acceptance of the paper. The pure language model tested in this work is a GPT-2 retrained by the authors on the same text corpus. And the other visual augmented models are all pretrained models from other papers. None of these comparisons can be perfectly fair on the training datasets. However, it seems that the model size is controlled, as this model uses a pretrained CLIP model during its training and is augmented by a large image database. If the training dataset can never be perfectly controlled, why not try models trained on much larger text corpus, such as the pretrained OPT models? Besides this, the numbers on these benchmarks seem to be also lower than the numbers I can find in other papers. For example, in the paper “Transferring Knowledge from Vision to Language: How to Achieve it and how to Measure it?” (about the Memory Color dataset), Table 4 shows numbers much higher than the numbers reported in this paper. Can the authors explain this difference?

Another minor issue about the evaluation is: what would be the current up-limit on these evaluation datasets from pure-language models? Have people tested the largest pretrained models on these datasets? Are these problems really hard for the models to resolve?

To help correctly evaluate the innovation of this work, can the authors also comment on how different this fusion layer proposed here is from the fusion layer in the Google Flamingo paper?

This is more of a question instead of an issue. It would be great to see how reducing the number of images in the auxiliary image database will influence the performance. I would imagine that having these images during the real-time inference makes the inference very slow, about which I cannot find any time estimation from the paper. So this could be an issue for real-world applications. To be clear, I don’t think this issue needs to be addressed in this work right now, but I want to get a sense of how annoying this is.

Finally, it would be good to know how the scaling will influence the results. Will large models like 350M make the performance better?


**Summary Of The Paper:**

This work proposes a novel architecture to do visual-augmented language modeling. Before each prediction of the next word, this architecture queries the most relevant images w.r.t. the current received part of the sentence using a pretrained CLIP model and a large image database. The authors trained this model on a large text corpus. They then showed that this model outperforms other models on multiple visual-language benchmarks, including Memory Color, Object Shape, Relative Size, and ColorTerms, by noticeable margins. The authors further show some ablation studies that the use of the image-retrieval module is helpful.

**Summary Of The Review:**

This work proposes a new architecture that achieves better performance than other models on the benchmarks they tested. The current version needs more justification about how real this improvement is and how innovative this architecture is.

---

> ### Author Response · Authors · 2022-11-17
> **Response to Reviewer j6aD (Part 1/2)**
>
> We thank you a lot for your detailed feedback and suggestions!
>
> Q1: Compare VaLM with OPT or other Large-Language Models.
>
> Thank you for the great suggestion on verifying the performance of large language models on the visual–knowledge intensive tasks for stronger and more fair comparisons. We evaluate the OPT (1.3B parameters) model on these visual–knowledge intensive tasks and the results are presented in the following Table. VaLM (124M parameters, k=8) significantly outperforms the OPT-1.3B on four visual-knowledge intensive datasets, which demonstrate both the challenge of resolving such visual-knowledge intensive tasks and the effectiveness of our method.
>
> | Model   | Memory Color | ColorTerms |   ObjectShape | RelativeSize     |
> |   ----  | ----  | ----  | ----   |  ----  |
> |  VaLM-124M |   58.64%  |     50.19%  |  59.41%  |  62.35%   |
> |  OPT-1.3B |  39.25%  |    41.03%   | 19.21% |  50.78%   |
>
> Q2: Difference with Memory Color Dataset Paper
>
> We did not observe a significant difference between our reimplementation of BERT-base on MemoryColor dataset and the result in the paper “Transferring Knowledge from Vision to Language: How to Achieve it and how to Measure it?”[1], which is 24.34% (our reimplementation in Table 2 Line 2) versus 25.2% ([1] Table 4 Line 4) . The Table 4 in paper [1] also presents the results of adopting unfiltered fine-tuning and filtered fine-tuning of BERT-base on MemoryColor dataset, which achieved much higher performance than the zero-shot evaluation. As we only evaluate VaLM in zero-shot manner without any task-specific tuning, it is not reasonable to compare zero-shot evaluation results to fine-tuning results.
>
> Q3: Difference from Flamingo Fusion Layer.
>
> VaLM is quite different from Google Flamingo in both architecture and motivation. Flamingo deploys the Gated-Cross Attention on the vision input X and language input Y in each layer of its model as a separate block. Each layer in Flamingo is more like the Transformer Encoder-Decoder architecture, in which the decoder layer contains both self-attention block and cross-attention block. The attention keys and values in such gated cross attention blocks are both X and the queries are Y. However, VaLM only adapts one layer in Transformer decoder architecture into a visual knowledge fusion layer. Our fusion layer has only one joint self-attention block to perform visual knowledge fusion. The keys and values of our joint self-attention mechanism are the concatenation of the visual input X and the context input Y, making the retrieved visual images like “auxiliary” context tokens in self-attention. Moreover, we treat each retrieval visual image encodings the same as the context token encodings, while Flamingo performs a complicated gating mechanism to decide the weighting between X and Y. We believe the fusion layer introduced in VaLM is more parameter-efficient and concise.

---

> ### Author Response · Authors · 2022-11-17
> **Response to Reviewer j6aD (Part 2/2)**
>
> Q4: The timecost brought by retrieval and imageset size effects.
>
>
> Thank you for your concern towards the timecost of the retrieval module. As for the problem of retrieval timecost, we discuss this topic in the Appendix A.2 with details on our experiments and potential measurements we can take to overcome such introduced timecost. As the introduced timecost is proportional to the size of imageset for dense retrieval, we provide more details on the relationship between retrieval timecost and imageset size, presented in the following Table.
>
> Deploying dense image retrieval increases the inference timecost to about 2.1 times of text-only GPT-2* baseline. This cost is negligible with larger-size language models because the model forward cost will increase many times while the retrieval cost will not change with the model size.
>
> Moreover, concluded from Table below, there is no significant performance decrease with the smaller imageset size from original 200M down to 10M. As 10M set is still large and sufficient for providing enough visual knowledge, we would consider deploying a 10M size imageset to train VaLM for potential real-time industry applications.
>
>
> | ImageSet Size   | Memory Color | ColorTerms |  ObjectShape |   RelativeSize |  Timecost (GPT2* as 1x)   |
> |   ----  |   ----  |  ----  | ----  |  ----  |    ----  |
> |          200M        |      53.99%    |   52.66% |       62.77%    |     85.03%        |         2.06x |
> |	100M	      |         53.50%   | 49.71% |    61.39% |    81.84%            |          1.91x |
> |          10M          |          51.79%  |   47.49% |  62.18% |        82.15%        |           1.79x  |
> |           1M           |          51.87%  |   46.31% |   48.51% |      82.35%         |             1.74x |
>
>
>
> Q5: The scaling effect.
>
> We trained the 355M model (GPT-2 Medium Size) of VaLM (k=8) and the results are significantly improved on all visual knowledge intensive benchmarks. The results are presented below and we would seek for more computation resources to train large size VaLM models.
>
> | Model Size   | Memory Color | ColorTerms |  ObjectShape | RelativeSize     |
> |   ----  | ----  |  ----  |  ----  | ----  |
> |  124M          |     58.64%     | 50.19%    | 59.41%  |  62.35%   |
> | 355M           |      65.82%  | 55.36%        |  70.0% |   72.79%    |

---

### Official Review · Reviewer_9KzG · 2022-10-25

**Confidence:** 4
**Correctness:** 3
**Technical Novelty And Significance:** 3
**Empirical Novelty And Significance:** 3
**Recommendation:** 6

**Clarity, Quality, Novelty And Reproducibility:**

Clarity: Good but some details are missing.
Quality: Good.
Novelty: Good.
Reproducibility: Since it requires a retrieving system on large-scale image datasets, reproducing might takes more effort.

**Strength And Weaknesses:**

Strengths:
1. The research problem of how to inject visual information into language modeling for visual-demand tasks is interesting. The method is intuitive and novel to a certain degree.
2. In many datasets, the performance of the proposed model outperforms compared methods/baselines

Weaknesses:
1. Indexing images, retrieving images and fusing image features to PLM all take extra computation and time as opposed to original PLM, especially retrieving I think. The analysis of additional training and inference time brought by the proposed method should be added.
2. Lack of experimental comparison against recent visual-augmented language model works. For example, Vokenization and iACE.
3. The database image quality matters for the retrieval and downstream task performance. But the ablation/analysis is missing. I mainly have two questions: (1) Does the dataset have to be an image-text dataset, such as ImageNet? Theoretically, it can also be an image-only dataset since CLIP is already trained. (2) Does the size of the database matter? How about we only take 100M, 10M, or even 1M data from Laion?
4. Some unclear illustrations. (1) In Sec2.3, is z_ij the image feature output from the same CLIP image encoder used in retrieval? (2) In the 3rd last sentence of Sec3.4, shouldn't it be "adopting image-specific bias outperforms directly sharing the bias"?

**Summary Of The Paper:**

To augment language models with relevant visual information, a visually-augmented language model (VALM) is proposed in this paper. The core idea is to retrieve relevant images by CLIP model and then fuse them to the second last Transformer layer of PLM.

**Summary Of The Review:**

My main concern is about the lack of experimental comparison with previous methods and the efficiency problem. So my initial rating is borderline reject.

---------- After rebuttal --------
My main concerns have been solved and I'd like to raise the score from borderline below to borderline above.

---

> ### Author Response · Authors · 2022-11-17
> **Response to Reviewer 9KzG**
>
> We thank you a lot for your detailed feedback! We will polish the expressions and revise the paper carefully.
>
> Q1: The timecost brought by retrieval and imageset size effects.
>
> Thank you for your concern towards the timecost of the retrieval module. As for the problem of retrieval timecost, we discuss this topic in the Appendix A.2 with details on our experiments and potential measurements we can take to overcome such introduced timecost. As the introduced timecost is proportional to the size of imageset for dense retrieval, we provide more details on the relationship between retrieval timecost and imageset size, presented in the following Table.
>
> Deploying dense image retrieval increases the inference timecost to about 2.1 times of text-only GPT-2* baseline. This cost is negligible with larger-size language models because the model forward cost will increase many times while the retrieval cost will not change with the model size.
>
> Moreover, concluded from the Table below, there is no significant performance decrease with the smaller imageset size from original 200M down to 10M. As 10M set is still large and sufficient for providing enough visual knowledge, we would consider deploying a 10M size imageset to train VaLM for potential real-time industry applications.
>
>
> | ImageSet Size   | Memory Color |  ColorTerms |   ObjectShape |   RelativeSize |  Timecost (GPT2* as 1x)   |
> |   ----  | ----  | ----  |    ----  |  ----  |      ----  |
> |          200M        |      53.99%    |     52.66% |     62.77%    |     85.03%        |         2.06x |
> |	100M	      |         53.50%   |  49.71% |  61.39% |    81.84%            |          1.91x |
> |          10M          |          51.79%  |  47.49% |  62.18% |        82.15%        |           1.79x |
> |           1M           |          51.87%  |    46.31% |  48.51% |      82.35%         |             1.74x |
>
>
>
> Q2: Comparisons with other visual-augmented works.
>
> We have compared VaLM with Vokenization on the visual-knowledge intensive benchmarks in our paper, and the results are shown in the Table below. iACE does not provide the public model checkpoints for quick reimplementation. We have requested that from the authors and will reproduce later. The results of Vokenization and the proposed model VaLM are presented in the Table below. VaLM (k=4) significantly outperforms Vokenization on 4 visual-knowledge intensive datasets.
>
> | Model  | Memory Color | ColorTerms | RelativeSize | ObjectShape |
> | ----  | ----  |   ----  |   ----   |      ----  |
> | VaLM(k=4) |  53.99% | 52.66% |  85.03% |            62.77%      |
> |   Vokenization     |	14.18%  |     19.97% |     48.35%             |            43.28% |
>
>
> Q3: Imageset Change.
>
> VaLM deploys LAION image dataset because LAION is one of the largest image dataset and the CLIP model we used for encoding images and text queries is also trained on LAION dataset. Our method does not need to use any image captions, thus deploying an image-only dataset like ImageNet as the retrieval image base would be definitely workable.
>
> Q4: Expressions.
>
> z_ij In Section 2.3 is the retrieved image feature output from the same CLIP image encoder used in retrieval, and we will make it more clear in later revisions. We will also polish the expression in the 3rd last sentence of Section 3.4 to help better understanding.

---

### Official Review · Reviewer_FCDh · 2022-10-25

**Confidence:** 4
**Correctness:** 4
**Technical Novelty And Significance:** 4
**Empirical Novelty And Significance:** 4
**Recommendation:** 10

**Clarity, Quality, Novelty And Reproducibility:**


Novelty and Significance: The idea is novel, timely and very relevant to recent progress in transformers. It provides a crisp solution to a clear problem. There are also many potential directions to both build on this work and apply it to improve current state of the art solutions.

Quality: The experiments provide strong support that VaLM works well, though further investigation of how the retrieval module is working would be good to add in this paper.

Clarity: The presentation is very simple and clear.

Reproducibility: The information in the paper is enough to implement the model. It provides hyperparameter details as well as the prompts used for evaluation. A code release is also promised.


**Strength And Weaknesses:**


Strengths
===

Incorporating visual background knowledge in a language only model makes a lot of sense and seems novel. Furthermore, VaLM is a simple and effective realization of the idea in practice.

The writing is clear and straightforward, without frills.



Weaknesses
===

There are couple areas where the model should have been evaluted:

* There is limited analysis of the images retrieved by the retrieval module. Using the figure 1 example, the idea is that it will say the sky is blue because the retrieved images tend to be blue. Are the retrieved images consistent with its answer? That is, given that it says the sky is blue, does it actually retrieve images where the sky is blue? If the retrieved images are manually substituted with alternative images where the sky is green then does VaLM say the sky is green? It would be good to have a systematic evaluation of the retrieved images, but it would also help to simply provide examples.

* A related concern is about the images recalled for each piece of a sentence. As I understand it, the visual knowledge fusion layer recalls a different set of K images for each sentence part (e.g. different images for "The color", "The color of", "The color of sky", etc.). How do the recalled images vary over the course of the sentence? Do they stay the same when non-semantic words like "of" are used? Do they change appropriately when semantic words like "sky" are used?

* There is no evaluation of how this impacts the model's runtime. How long does a forward pass take when generating every word requires a kNN lookup?


There are also some points where the presentation could be clearer:

* The motivation in 3.2 ("object properties rarely appears in text corpora") seems like it should also belong in the introduction. To me this is a key reason to expect this approach to be helpful.

* The text doesn't mention what the "Majority" row in Table 4 means.


Finally, here are some minor suggestions and comments:

* The attention mechanism over K retrieved images is novel, but a fairly straightforward extension of the attention mechanism once one has already decided to retrieve relevant images. The major contribution is in retrieving relevant images.

* Can VaLM be used as a vision-language model? Essentially, what if it was given vision and langauge tasks (e.g., VQA) and then the image retrieval module was replaced with a module that simply returned the image(s) associated with the VL task example? Would it perform well at the VL task? In general, does it treat the retrieved images as global / abstract context or does it also consider them as specific context local to those images?

**Summary Of The Paper:**


This paper improves a language model's performance on pure language tasks about visual concepts by augmenting its internal representation with dynamically retrieved images.

(motivation)
Global context (external knowledge about entities, relations, etc.) has been incorporated into pre-trained language models (PLMs), but so far it has not been visual. This paper incorporates external visual knowledge into their PLM (called VaLM) to
1) prevent VaLM from hallucinating inconsistent statements and
2) give VaLM visual knowledge that is harder to obtain from text.

(approach)
VaLM is a standarder decoder style PLM, used like any standard PLM and trained on the CC-100 dataset (text only). The difference is that the second to last self attention layer is replaced with a Visual Knowledge Fusion Layer (VKFL). First the VKFL uses CLIP to retrieve K (4 or 8) relevant images from a database of 200M natural images (from LAION) where relevance is according to the text input VaLM has seen so far. Second the VKFL fuses these relevant images with VaLM's hidden state at the previous layer to produce the output hidden state - the tokens are the same, they just have additional visual knowledge.

(experiments)
Results show that
1. When asked to complete text prompts about the typical color, shape, and relative size of objects, VaLM outperforms both PLMs and pretrained vision-language models by a large margin.
2. VaLM outperforms baselines at pysical commensense QA (PIQA).
3. VaLM is at parity with and sometimes better than baselines at traditional language understanding and language modeling tasks.
4. Ablations show disabling image retrieval or retrieving random (not relevant) images hurts performance.
5. Ablations also justify design decisions in the attention mechanism.

VaLM improves language modeling performance by adding global image context internally.

**Summary Of The Review:**


The novelty, potential significance, quality, clarity, and potential reproducibility of this work are all high, so it is a very strong submission.

---

> ### Author Response · Authors · 2022-11-17
> **Response to Reviewer FCDh**
>
> We would like to thank you for your time and constructive suggestions. We will polish the expressions and revise the paper carefully.
>
>
> Q1:  Retrieved image quality and substitution effects.
>
> We provide some retrieval visualization samples in Appendix A.1 Figure 2 in the supplementary material. For the query prompt “The color of sky is ”, we list the top-4 retrieved images in Figure 2.(b), all of which are images of blue sky and can be useful augmentations for language modeling. If we manually substitute the retrieved images as green sky, the answer of the sky color will not change to green because the prediction is based on both visual knowledge in retrieved images and language knowledge from language model itself. If the retrieved visual knowledge is far from the commonsense, our model seems to be able to resolve it by taking the language commonsense knowledge as the answer.
>
>
> Q2: Retrieval image for different contexts in a sentence.
>
> We visualize the retrieved image sets for three different contexts in a sentence “the color”, “the color of”, and “the color of sky”. The retrieved images for the contexts “the color” and “the color of” lack meaningful and organized visual knowledge. The retrieved images for “the color” are mainly signs or icons with the word “color” in it. The retrieved images for “the color of” are mainly color palettes. But when the visually salient word “sky” is appended, the 8 retrieved images become useful and are 5 images of blue sky, 2 images of pink sky with some kind of evening glow, and 1 image of black and green sky with aurora. Such retrieved images for the context with visually salient words “the color of sky” would be helpful for the language model to learn and gain visual knowledge through fusion.
>
>
> Q3: The timecost brought by retrieval and imageset size effects.
>
> Thank you for your concern towards the timecost of the retrieval module. As for the problem of retrieval timecost, we discuss this topic in the Appendix A.2 with details on our experiments and potential measurements we can take to overcome such introduced timecost. As the introduced timecost is proportional to the size of imageset for dense retrieval, we provide more details on the relationship between retrieval timecost and imageset size, presented in the following Table.
>
> Deploying dense image retrieval increases the inference timecost to about 2.1 times of text-only GPT-2* baseline. This cost is negligible with larger-size language models because the model forward cost will increase many times while the retrieval cost will not change with the model size.
>
> Moreover, concluded from the Table below, there is no significant performance decrease with the smaller imageset size from original 200M down to 10M. As 10M set is still large and sufficient for providing enough visual knowledge, we would consider deploying a 10M size imageset to train VaLM for potential real-time industry applications.
>
>
>
> | ImageSet Size   | Memory Color |  ColorTerms |   ObjectShape |   RelativeSize |  Timecost (GPT2* as 1x)   |
> |   ----  | ----  | ----  |    ----  |  ----  |      ----  |
> |          200M        |      53.99%    |     52.66% |     62.77%    |     85.03%        |         2.06x |
> |	100M	      |         53.50%   |  49.71% |  61.39% |    81.84%            |          1.91x |
> |          10M          |          51.79%  |  47.49% |  62.18% |        82.15%        |           1.79x |
> |           1M           |          51.87%  |    46.31% |  48.51% |      82.35%         |             1.74x |
>
>
> Q4:  Some Expressions.
>
> We will move the motivation sentence in Section 3.2 to the introduction section to help better demonstrate the importance of the problem and the approach. The “majority” in Table 4 of the paper means that we choose the class with most samples as the predicted classes for all test samples and compute the accuracy based on that. We will make it clear in our revision.

---

### Official Review · Reviewer_f9V4 · 2022-10-26

**Confidence:** 4
**Correctness:** 4
**Technical Novelty And Significance:** 3
**Empirical Novelty And Significance:** 3
**Recommendation:** 6

**Clarity, Quality, Novelty And Reproducibility:**

The paper is quite clear, quality of writing and results is high. The method is novel and authors mention they will be releasing the code on github later.

**Strength And Weaknesses:**

**Strengths**:

-- The empirical results in the paper are quite strong and surpass pre-trained text models (GPT) as well as vision-language models (VisualBERT) on 4 reasoning task datasets and several language understanding/modeling tasks. While the differences in language-only tasks is small, the gains compared to baselines in visual reasoning tasks is large as claimed.

-- The proposed model (esp visual knowledge fusion component) is novel and elegant and simple, which will serve as motivation for following works.

-- The paper is clearly written and easy to follow.

**Weaknesses**:

-- The main weakness in the method is the use of frozen image retrieval component. The concern is that if this component is not end-to-end trained with the rest of the model, will the model quality be limited by the quality of image retrieval. While the reviewer acknowledges that empirical results show a large gap between CLIP and VALM on all tasks, it is worth wondering if the gains are due to embedding multiple images with text in VALM (compared to single image-text pair used originally in CLIP). An ablation with different values of k (number of images retrieved) will be helpful here.

**Summary Of The Paper:**

This paper proposes a pre-training framework, called VALM, to jointly train on image-text data. The novelty of this work, compared to previous works in similar field, is how the image-text pairs are created. While previous works use pre-curated image-text aligned pairs, this work instead uses images retrieved using text as a query and then jointly fuses them through attention layers. The claim is that this will help the model perform better on tasks requiring visual commonsense reasoning.

**Summary Of The Review:**

Overall, quite an interesting and novel idea in the space of vision-language models. The claims are verified by empirical results. VALM outperforming on tasks compared to other vision-language models is a strong result. The weakness related to understanding why this method works better than other vision-language models and whether frozen image retrieval will limit the method's effectiveness is something the reviewer would like to understand and engage with the authors further.

---

> ### Author Response · Authors · 2022-11-17
> **Response to Reviewer f9V4**
>
> Thanks for suggesting a closer comparison with CLIP. We further conduct another ablation study by setting the number of augmented images k=1 for VaLM, which is very similar to the CLIP inference. The results are presented in the following Table. VaLM (k=1) significantly outperforms CLIP in all visual-knowledge intensive tasks, validating the effectiveness of our method.
>
>
>
> | Model |   k |  Memory Color | ColorTerms | ObjectShape  | RelativeSize |
> |   ----  | ----  | ----  | ----  |----  |----  |
> |VaLM | 1            |     52.43%    | 46.01% |  64.55%    |      86.73% |
> |VaLM | 4           |   53.99%   | 52.66%    |     62.77%|            85.03%   |
> |VaLM |  8           |     58.64%   | 50.19%  | 59.41%  |               62.35% |
> | CLIP| N/A         |  26.25%  |   23.08% |   13.66%    | 47.99%    |

---

### Decision · Program_Chairs · 2023-01-20

**Decision:**

Accept: poster

**Justification For Why Not Higher Score:**

I think this is a solid paper, but the tasks are mostly vision + language tasks, the method is expensive and relies on a good retriever, making the applications of such model limited.

**Justification For Why Not Lower Score:**

n/a

**Metareview: Summary, Strengths And Weaknesses:**

The paper proposes a pre-training framework to augment the visual information to the text pretrained models. The method is based on a an image retrieval module given a textual context and a fusion layer to comsume the visual information. The experiments are carefully done and compared with essential baselines. The reviewers are generally happy about the paper. Although reviewer 9KzG did not raise the score in the end, I feel his concerns are addressed in the rebuttal stage. I therefore would recommend acceptance of this paper to ICLR.

**Note From Pc:**

if the above contains the word "oral" or "spotlight" please see: "oral" presentation means -> notable-top-5% and "spotlight" means -> notable-top-25%. As stated in our emails, we are disassociating presentation type from AC recommendations